# Ta_3_N_5_ Nanobelt-Loaded Ru Nanoparticle Hybrids’ Electrocatalysis for Hydrogen Evolution in Alkaline Media

**DOI:** 10.3390/molecules28031100

**Published:** 2023-01-21

**Authors:** Xinyu Zhang, Lulu Xu, Xingcai Wu, Yourong Tao, Weiwei Xiong

**Affiliations:** 1Key Laboratory of Mesoscopic Chemistry of MOE, State Key Laboratory of Coordination Chemistry, School of Chemistry and Chemical Engineering, Nanjing University, Nanjing 210093, China; 2School of Environmental & Chemical Engineering, Jiangsu University of Science and Technology, Jiangsu 212003, China

**Keywords:** tantalum nitride, ruthenium, hydrogen evolution, theoretical calculations

## Abstract

Electrochemical hydrogen evolution is a highly efficient way to produce hydrogen, but since it is limited by high-cost electrocatalysts, the preparation of high-efficiency electrocatalysts with fewer or free noble metals is important. Here, Ta_3_N_5_ nanobelt (NB)-loaded Ru nanoparticle (NP) hybrids with various ratios, including 1~10 wt% Ru/Ta_3_N_5_, are constructed to electrocatalyze water splitting for a hydrogen evolution reaction (HER) in alkaline media. The results show that 5 wt% Ru/Ta_3_N_5_ NBs have good HER properties with an overpotential of 64.6 mV, a Tafel slope of 84.92 mV/dec at 10 mA/cm^2^ in 1 M of KOH solution, and good stability. The overpotential of the HER is lower than that of Pt/C (20 wt%) at current densities of 26.3 mA/cm^2^ or more. The morphologies and structures of the materials are characterized by scanning electron microscopy and high-resolution transmission electron microscopy, respectively. X-ray photoelectron energy spectroscopy (XPS) demonstrates that a good HER performance is generated by the synergistic effect and electronic transfer of Ru to Ta_3_N_5_. Our electrochemical analyses and theoretical calculations indicate that Ru/Ta_3_N_5_ interfaces play an important role as real active sites.

## 1. Introduction

Hydrogen is a high-energy-density and environmentally friendly gas. Its only combustion product is pollution-free water, so hydrogen is considered an ideal alternative to traditional fossil fuels and is important in future energy systems [1,2]. Research demonstrates that water electrolysis [3,4,5,6] and photoelectrolysis [7,8] are very effective methods of hydrogen production, which can convert solar and wind energies into hydrogen fuel [9,10]. However, large-scale electrochemical hydrogen evolution is still a challenge because it is limited by high-cost and inefficient electrocatalysts. Water electrolysis is usually carried out in alkaline or acid aqueous solution to improve conductivity. Because of its suitable manufacture safety and stable output, alkaline hydrogen evolution reaction (HER) is considered an important cathode reaction, with application potential in the chlor-alkali industry and anion exchange membrane (AEM)-based water electrolysis. Compared with acidic HER, alkaline HER can use various non-precious metal materials for electrocatalysis, such as transition metal oxides, phosphides, carbides, sulfides, and so on [11,12,13,14]. Nevertheless, alkaline HER is generally rather inactive at a high current density due to the low proton concentration. The sluggish kinetics of alkaline HER derived from extra water dissociation can result in higher overpotentials and reaction rates 2–3 orders of magnitude lower than those in the acid condition [15]. Therefore, developing high-efficiency and low-cost electrocatalysts to accelerate the alkaline HER kinetics is very important for water electrolysis.

Platinum (Pt) is considered the best HER electrocatalyst due to its optimal hydrogen adsorption energy [16,17], but it cannot yet be applied large-scale because it is rare in nature. Ruthenium (Ru) has a higher natural abundance than Pt, and its price is only 1/25 that of Pt [18,19]. Its hydrogen binding energy (65 Kcal mol^−1^) is slightly higher than that of Pt (62 Kcal mol^−1^) to increase the energy of hydrogen desorption, to prevent the regeneration of active sites [18]. If Ru is modified, alloyed, or doped, its HER catalytic activity can visibly improves [20,21,22,23]. For example, Sun et al. prepared a B-Ru@CNT (carbon nanotube) electrocatalyst with excellent HER activity, yielding overpotentials of 17 and 62 mV at a current density of 10 mA cm^−2^ in alkaline and acidic solutions, respectively. These experimental and theoretical results indicate that the incorporation of B weakens the Ru–H bond and downshifts the d-bond center of Ru from the Fermi level by reducing the electron density of Ru [24]. Similarly, the 3 wt% Ru/CeO_2_ electrocatalyst also demonstrates excellent HER performance, which confirms the downshift of the d-band of Ru, and the reduction of the Ru–H bond energy [25]. Therefore, Ru is an up and coming material for electrocatalytic HER. Ta_3_N_5_ is a red powder with an octahedral structure and a narrow band gap of about 2.08 eV, which has good photosensitive properties [26] and is widely applied in solar-driven photocatalytic or photoelectrochemical (PEC) water splitting [27,28]. Its analog TaON is also a good photosensitive agent [29] and an ideal photoelectro-catalyst for hydrogen evolution [30,31]. This begs the question, if Ru/Ta_3_N_5_ is used in electrocatalytical HER under a non-solar-driven condition, how do its properties support HER?

Here, Ta_3_N_5_ nanobelts (NBs) are used as a supporter of Ru nanoparticles to construct electrocatalysts for water splitting in alkaline media. The results show that the hydrogen evolution performance of 5 wt% Ru/Ta_3_N_5_ NBs is comparable to 20 wt% Pt/C. This excellent performance is due to the valence electrons on the Ru surface transferring to Ta_3_N_5_, which favors the dissociation of water and reduces the adsorption energy of hydrogen.

## 2. Results

### 2.1. Structures and Morphologies of Catalysts

To prepare Ru/Ta_3_N_5_ NBs, we synthesized TaS_3_ NBs and then converted them to Ta_3_N_5_ NBs in NH_3_ atmospheres. Subsequently, various quantities of Ru^3+^ were adsorbed on the surface of Ta_3_N_5_ NBs by the dipping method, before Ru was loaded on the Ta_3_N_5_ NBs by reduction. The structures and morphologies of the catalysts were characterized by X-ray diffractometry (XRD) and scanning electron microscopy (SEM), respectively.

Figure 1a–d show the SEM images of TaS_3_ NBs, Ta_3_N_5_ NBs, 5 wt% Ru/Ta_3_N_5_ NBs, and 5 wt% Ru/TaS_3_ NBs, respectively, which reveal that TaS_3_ NBs are nanobelts with a width of 1–4 μm, a thickness of about 10–200 nm, and a length of up to a centimeter, and that the sizes of Ta_3_N_5_ NBs are close to those of TaS_3_ NBs. Because Ru nanoparticles (NPs) were loaded on the surface of Ta_3_N_5_ and TaS_3_ NBs by reduction of Ru^3+^ in an aqueous solution, the lengths of Ta_3_N_5_ and TaS_3_ NBs were diminished. The sizes of Ru NPs loaded were about 3–15 nm. Figure 1e shows the XRD patterns of TaS_3_ NBs, Ta_3_N_5_ NBs, 5 wt% Ru/Ta_3_N_5_ NBs, and 5 wt% Ru/TaS_3_ NBs. TaS_3_ and Ta_3_N_5_ NBs were assigned to orthogonal phases of TaS_3_ (JCPDS no. 18-1313) and Ta_3_N_5_ (JCPDS no. 79-1533), respectively [26]. No Ru diffraction peaks can be observed in the XRD profiles of 5 wt% Ru/TaS_3_ and 5 wt% Ru/Ta_3_N_5_, which may be attributed to the small size and low load of Ru.

The Ta, N, and Ru elements in the 5 wt% Ru/Ta_3_N_5_ NBs are confirmed by the energy dispersive X-ray spectroscopy (EDX) shown in Figure 1f, and the mass ratio (%) of Ta, N and Ru is 83.98:14.00:2.02, demonstrating that the percentage of Ru is lower than 5 wt% (real value), although it is possible that the EDX analysis is only approximate.

Figure 2a,b present the high-resolution transmission electron microscopy (HRTEM) images of 5 wt% Ru/Ta_3_N_5_ NBs, and it is evident that the addition of Ru does not alter the morphology of Ta_3_N_5_. The Ru NPs with a size of about 3–15 nm can be observed on Ta_3_N_5_. As shown in Figure 2a,b, lattice stripe spacing of 0.22 nm and 0.50 nm belong to the (100) plane of Ru and the (002) plane of Ta_3_N_5_, respectively.

To find the optimal load for Ru, we prepared Ru/Ta_3_N_5_ NB catalysts with different mass fractions of Ru (1 wt%, 3 wt%, 7 wt%, and 10 wt%). Figure 3a–d show SEM images of those, respectively, and the XRD patterns are shown in Appendix A, which are indexed as orthogonal Ta_3_N_5_ (JCPDS no. 79-1533). Because Ru NPs are too little or the strong peak (2θ: 44°) of Ru overlaps with that of Ta_3_N_5_, it is difficult to observe the Ru peak.

### 2.2. The Surface Properties

Figure 4a–d present the X-ray photoelectron energy spectroscopy (XPS) of Ru NPs, Ta_3_N_5_ NBs, and 5 wt% Ru/Ta_3_N_5_ NBs. Figure 4a shows the XPS survey spectra of those, which further confirm the existence of Ru, N, and Ta in 5 wt% Ru/Ta_3_N_5_ NBs. As shown in Figure 4b, the peaks at 396.23 eV and 403.28 eV of 5 wt% Ru/Ta_3_N_5_ NBs correspond to the binding energies (BEs) of N 1s and Ta 4p_3/2_, respectively, while the peaks at 395.89 eV and 403.11 eV of Ta_3_N_5_ NBs correspond to the BEs of N 1s and Ta 4p_3/2_ [32,33], respectively. Figure 4c presents the fine XPS spectra of Ta 4f of the Ta_3_N_5_ and 5 wt% Ru /Ta_3_N_5_ NBs. The two peaks at 24.21 and 26.17 eV of 5% Ru/Ta_3_N_5_ correspond to the BEs of Ta 4f_7/2_ and Ta 4f_5/2_, respectively, while the two peaks at 24.24 eV and 26.16 eV correspond to the BEs of Ta 4f_7/2_ and Ta 4f_5/2_ of Ta_3_N_5_, respectively.

Figure 4d presents the Ru 3d_5/2_ spectra of 5 wt% Ru /C and 5% Ru/Ta_3_N_5_ NBs, where the peaks at 280.30 and 281.46 eV correspond to their Ru3d_5/2_ BEs, respectively, and the bias is 1.2 eV. The increase in the Ru 3d_5/2_ BEs from 5 wt% Ru/C to 5 wt% Ru/Ta_3_N_5_NBs demonstrates that the electrons of Ru transfer to Ta_3_N_5_ on the surface of 5 wt% Ru/Ta_3_N_5_NBs so that the adsorption energy of H on Ru lowers, which is helpful to increase the electrocatalytic activity for HER.

### 2.3. Hydrogen Evolution Performance

Figure 5a,b show the linear scanning voltammetry (LSV) and Tafel curves of 5 wt% Ru/Ta_3_N_5_ NBs, Ta_3_N_5_ NBs, 5 wt% Ru/TaS_3_ NBs, Ru (5 wt% Ru/C, C: conductive carbon), and 20 wt% Pt/C in 1M of nitrogen-saturated KOH solution at a rotating rate of 1600 rpm and a scanning rate of 5 mV/s.

Their overpotentials are 64.6, 394.6, 261.6, 96.6, and 49 mV at a current density of 10 mA/cm^2^, respectively, while their Tafel slopes are 84.92, 135.98, 139.24, 123.04, and 59.9 mV/dec, respectively. Clearly, the activity of 5 wt% Ru/Ta_3_N_5_ NBs for HER is superior to those of the other three, except for 20 wt% Pt/C at 10 mA/cm^2^, but when the current density is greater than 26.3 mA/cm^2^, the activity of 5 wt% Ru/Ta_3_N_5_ NBs is superior to 20 wt% Pt/C. Because the HER properties of 5 wt% Ru/Ta_3_N_5_ NBs are superior to 5 wt% Ru/ TaS_3_ NBs, Ta_3_N_5_ NBs as supporters are better than TaS_3_ NBs.

In order to compare the effects of different Ru-loaded quantities on HER, we experimented on Ru/Ta_3_N_5_ NBs with various ratios. Figure 5c,d demonstrate the LSV and Tafel curves of 1 wt%, 3 wt%, 5 wt%, 7 wt%, and 10 wt% Ru/Ta_3_N_5_ NBs, respectively, where their overpotentials are 176.6, 154.6, 64.6, 214.6, and 220.6 mV at a current density of 10 mA/cm^2^, respectively, and their Tafel slopes are 119.38, 109.72, 84.92, 138.31, and 135.08 mV/dec, respectively, so 5 wt% for Ru/Ta_3_N_5_ NBs is the optimal ratio. Because 5 wt% for Ru/Ta_3_N_5_ NBs has the gentlest Tafel slope, this reveals it has favorable electrocatalytic kinetics with a high transfer coefficient.

To explain the difference in the catalytic activities, the C_dl_ (double-layer capacitance) of 5 wt% Ru/Ta_3_N_5_ NBs, Ta_3_N_5_ NBs, and 5 wt% Ru/TaS_3_ NBs was measured as 7.28, 3.74, and 1.94 mF/cm^2^, respectively, as shown in Figure 5e, and the CV curves at 1.1–1.35 V are shown in Appendix A. The C_dl_ values of 1 wt%, 3 wt%, 5 wt%, 7 wt%, and 10 wt% Ru/Ta_3_N_5_ NBs were measured as 4.12, 3.99, 7.28, 3.81, and 3.54 mF/cm^2^, respectively, as shown in Figure 5f, and the CV curves at 1.1–1.35 V are shown in Appendix A. Hence, the good catalytic activity of 5 wt% Ru/Ta_3_N_5_ NBs can be attributed to their high electrochemical activity surface area (ECSA).

Electrochemical impedance spectroscopy (EIS) reveals the interfacial properties between the electrolyte and the electrode, and the semicircle in the high-frequency region (Nyquist plot) represents the charge transfer process. Figure 6a shows the Nyquist plots of 5 wt% Ru/Ta_3_N_5_ NBs, Ta_3_N_5_ NBs, 5 wt% Ru/TaS_3_ NBs, and Ru NPs (5% Ru/C) measured in 1 M KOH of nitrogen saturation, where Z′ and Z′′ are real and imaginary parts of the impedance, respectively. The NBs’ and NPs’ transfer resistances (R_ct_) are 9.721, 4621, 73.51, and 72.78 ohm, respectively. Figure 6b shows the Nyquist plots of 1 wt%, 3 wt%, 5 wt%, 7 wt%, and 10 wt% Ru/Ta_3_N_5_NBs measured in 1 M KOH of nitrogen saturation. Their transfer resistances (R_ct_) are 161.7, 77.56, 9.721, 71.07, and 32.35 ohm, respectively. The equivalent circuit of the Nyquist plots fitted is shown in the inset of Figure 6a. Because the transfer resistance of 5 wt% Ru/Ta_3_N_5_ NBs is minimal, the electron transfer rate of HER increases, which is helpful to improve the electrocatalytic activity. Figure 6c shows the chronoamperometric curves of the 5% Ru/Ta_3_N_5_ NBs coated on 0.5 cm × 0.5 cm foam nickel at 10 and 40 mA/cm^2^ for 25 h (the slurry is the same as that used in GCE), which confirms the good hydrogen evolution stability of the materials. For comparison, a few results for Ru are included in Table 1.

In order to theoretically inspect the enhanced effect of interfaces on the distinguished HER, we also conducted density functional calculations on the adsorption of H^+^ on the surface and interface of Ru/Ta_3_N_5_. The adsorption free energy of hydrogen (ΔG_H*_) on the catalyst surface is recognized as the main descriptor to evaluate the HER activity in both acidic and alkaline solutions. An excellent HER catalyst requires an optimal ΔG _H*_ value close to zero or that of Pt. The ΔG_H*_ values of the H adsorption on the different sites of Ru, Ta_3_N_5_, and Ru/Ta_3_N_5_ are shown in Figure 6d, and corresponding adsorption structures are shown in Figure 7a–f. By comparison, the ΔG_H*_ of Ta_3_N_5_–Ru–H4 reveals that the adsorption structure, as shown in Figure 7f, is the least developed (−0.104 eV) as it is close to zero, so the site of the Ru atom near the interface is the best site of H adsorption.

## 3. Materials and Methods

### 3.1. Experimental Reagents

Tantalum (Ta) plates (99.5%), sulfur powders (99.99%), iodine powders (99.8%), RuCl_3_·H_2_O (98%), and NaBH_4_ (98%) were bought from China National Chemical Reagent Co., Ltd. NH_3_ (99%) was bought from Nanjing Tianze Gas Co., Ltd., Nanjing, China.

### 3.2. Preparation of the Catalysts

The prepared procedures were divided into three steps. Firstly, a 524.5 mg Ta plate (5 mm × 0.2 mm × 30 mm), sulfur powder (Ta: S molar ratio = 2:1), and 10 mg of iodine were weighed, placed in quartz tubes (Φ0.6 cm × 10 cm), and sealed by a hydrogen torch in a 10^−2^ Pa vacuum. The sealed quartz tube was placed in a horizontal furnace (Φ 4 cm × 32 cm) with a temperature gradient from the center to both sides of above 10 °C/cm, where the reagent was in the center of the furnace. The furnace was heated from room temperature to 550 °C at 10 °C/min, and maintained for 6 h, then naturally cooled to room temperature before TaS_3_ NBs were scraped from Ta substrate. Secondly, the TaS_3_ NBs were placed in a ceramic boat and put in the center of the horizontal furnace, heated from room temperature to 850 °C in flowing NH_3_ (100 mL/min), and maintained at 850 °C for 3 h, then naturally cooled to room temperature so that Ta_3_N_5_ NBs were obtained. Thirdly, 20 mg Ta_3_N_5_ NBs were added to a round-bottomed flask with 10 mL of ethanol and 20 mL of deionized water, then stirred to form a uniform suspension. Subsequently, 1.117 mL of 2 mg/mL RuCl_3_·H_2_O solution was injected and stirring continued for 12 h before 1.0 mL of NaBH_4_ aqueous solution (NaBH4:Ru = 3:1 mol/mol) was slowly added to the above system, and stirring resumed for 1 h. Finally, the products were separated by multiple centrifugation and washed with deionized water and ethanol. The collected products were dried by vacuuming at 60 °C to form the 5 wt% Ru/Ta_3_N_5_ NBs catalyst, and 5 wt% Ru/C and 5 wt% Ru/TaS_3_ NBs were similarly prepared. Ru/Ta_3_N_5_ NBs with different percentages of Ru and Ta_3_N_5_ NBs (1, 3, 7 and 10 wt%) were also prepared by adjusting the ratio of Ru and Ta_3_N_5_ in the NBs.

### 3.3. Characterization

The structure and morphology of the products were characterized by a Rigaku D/Max-rC diffractometer (XRD) with monochromatized Cu K_α1_-radiation (λ = 1.5406 Å), a S-4800 scanning electron microscope (SEM) with an energy dispersive X-ray spectrometer (EDX), and a JEOL-JEM-2010 high-resolution transmission electron microscope (HRTEM) with image and selected area electron diffraction (SAED). The valence states of the surface elements were analyzed by X-ray photoelectron energy spectroscopy (XPS, PHI 5000 Versa Probe, A1 K_α_ radiation, hν =1486.6 eV, Japan), and the binding energy was corrected based on C1s = 284.6 eV.

### 3.4. Electrochemical Performance Test

The 8 mg catalyst and 0.5 mg of carbon black (Vulcan XC-72) were dispersed in 500 μL of aqueous solution and sonicated for 20 min to disperse into uniform ink. Then, 5 μL of catalyst ink and 3 μL of Nafion were injected into a polished glass carbon electrode (GCE, surface area 0.1256 cm^2^) to naturally dry. Through the same method, the 20 wt% Pt/C catalyst was coated on GCE. All electrochemical measurements were performed by a three-electrode system at the electrochemical workstation (CHI 760E, Chenhua, China). The three-electrode system was formed with the carbon rod as the counter electrode, Hg/HgO as the reference electrode, and the rotating disk electrode with GCE as the working electrode. Electrochemical tests were performed in 1 M of KOH solution at room temperature. The electrocatalytic properties for hydrogen evolution under an alkaline solution were determined by linear scanning voltammetry (LSV, 5 mV/s, negative sweep), circulating voltammetry (CV, 50 mV/s, negative sweep), electrochemical impedance spectrum (EIS), and chronoamperometry methods. The electrochemical activity surface areas (ECSAs) of catalysts were measured by the double-layer capacitance (C_dl_) method because C_dl_ is proportional to ECSA. ECSA = C_dl_/C_s_, where C_dl_ is the double-layer capacitance and C_s_ is the specific capacitance of a planar surface that is atomically smooth under identical electrolyte conditions (C_s_ = 40 μF·cm^−2^ in this case) [34]. The C_dl_ values were measured by the CV curves at 1.1–1.35 V and by changing the scanning speed (10–100 mV/s). C_dl_ = (j_a_ − j_c_)/(2v) = Δj/(2v), where j_a_ and j_c_ are the anodic and cathodic voltametric current densities, respectively, and v is the scan rate. The EIS was tested at a frequency of 10^4^–10^−2^ Hz, amplitude of 5 mV, and rotational speed of 1600 rpm. All the potential values measured were corrected by the reversible hydrogen electrode potential (RHE): E_RHE_ = E_Hg/HgO_ + 0.0591 pH + 0.095 V.

### 3.5. Theoretical Computation

The Dmol^3^ module in the Materials Studio 8.0 package was employed to perform the spin polarization DFT calculations, and the exchange–correlation energy is depicted by the functional of generalized gradient approximation in the Perdew–Becke–Ernzerhof (PBE) form. The double numerical plus polarization (DNP) basis set was adopted, while accurate DFT Semi-Core Pseudopots (DSPP) was used for the metal atoms. All models were calculated in periodical boxes with a vacuum slab of 15 Å to separate the interaction between periodic images [35]. More details of the DFT calculations can be found in Appendix A.

## 4. Conclusions

In this project, we studied the electrocatalytic hydrogen evolution performance of 1~10% Ru/Ta_3_N_5_ NBs, Ta_3_N_5_ NBs, and 5 wt% Ru/TaS_3_ NBs in alkaline media. We discovered that 5 wt% Ru/Ta_3_N_5_NBs had good HER properties with an overpotential of 64.6 mV and a Tafel slope of 84.92 mV/dec at 10 mA/cm^2^. The good HER performance of 5 wt% Ru/Ta_3_N_5_ stems from the synergistic effect and electronic transfer of Ru to Ta_3_N_5_. This article provides an efficient method for designing low-cost and efficient transition Ru-based electrocatalysts for HER.

## Figures and Tables

**Figure 1 molecules-28-01100-f001:**
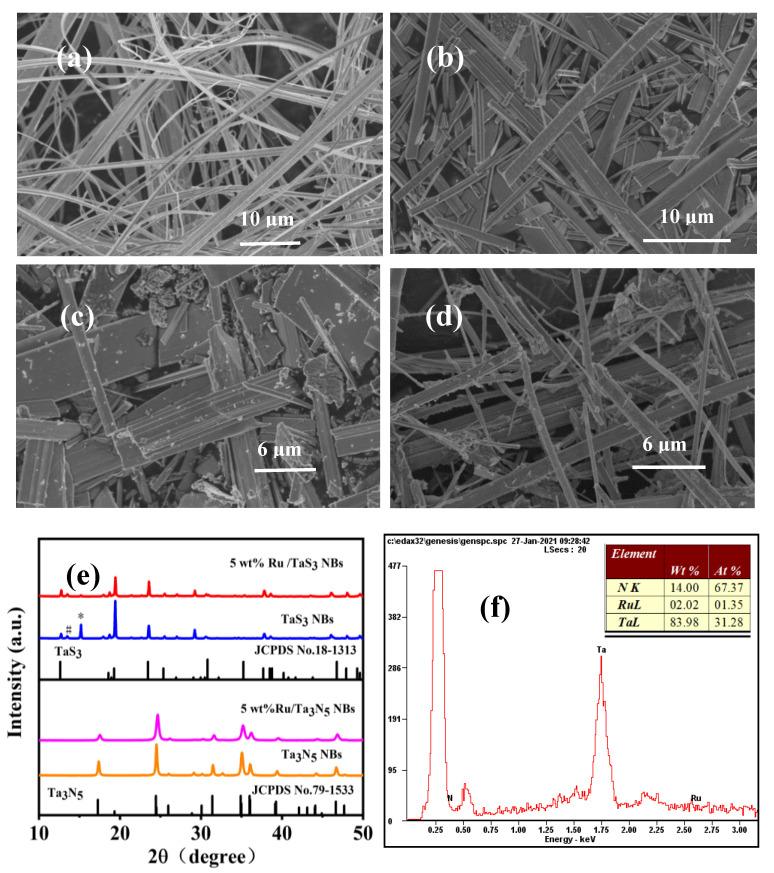
SEM images of (**a**) TaS_3_ NBs, (**b**)Ta_3_N_5_ NBs, (**c**) 5 wt% Ru/Ta_3_N_5_ NBs, and (**d**) 5 wt% Ru/TaS_3_ NBs; (**e**) XRD patterns of TaS_3_ NBs, Ta_3_N_5_ NBs, and 5 wt% Ru/Ta_3_N_5_ NBs (* and # are unknown peaks); (**f**) EDX spectrum of 5 wt% Ru/Ta_3_N_5_ NBs (inset: chemical components).

**Figure 2 molecules-28-01100-f002:**
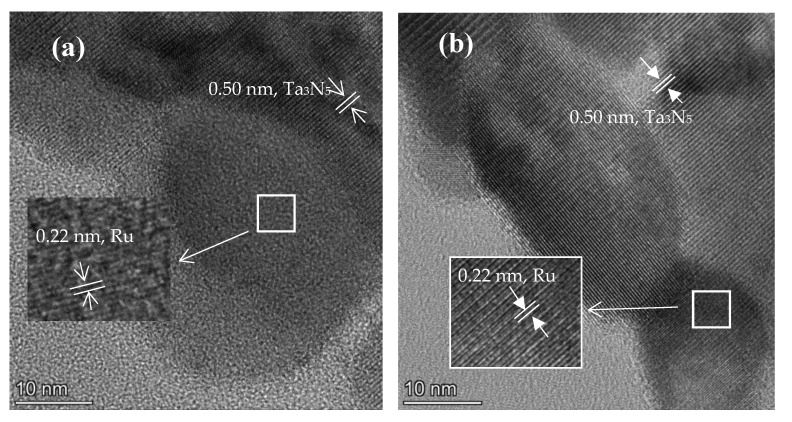
(**a**,**b**) HRTEM images of 5 wt% Ru/Ta_3_N_5_ NBs.

**Figure 3 molecules-28-01100-f003:**
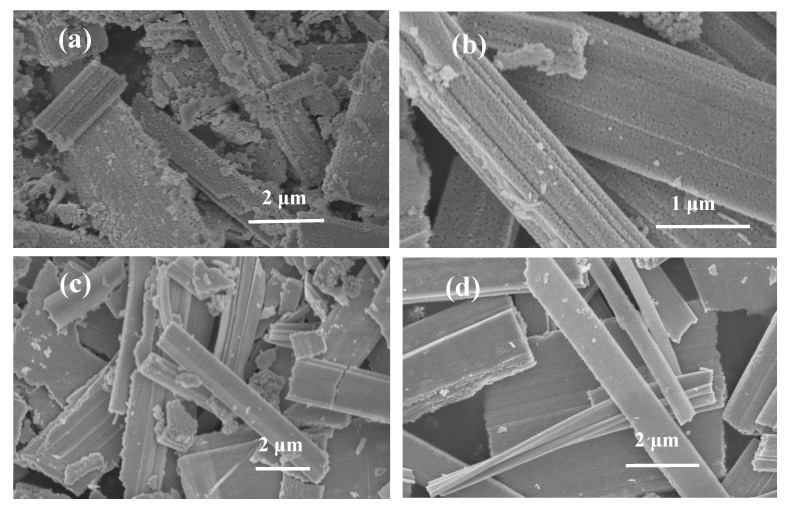
SEM images of (**a**) 1 wt% Ru/Ta_3_N_5_ NBs, (**b**) 3 wt% Ru/Ta_3_N_5_ NBs, (**c**) 7 wt% Ru/Ta_3_N_5_ NBs, and (**d**) 10 wt% Ru/Ta_3_N_5_ NBs.

**Figure 4 molecules-28-01100-f004:**
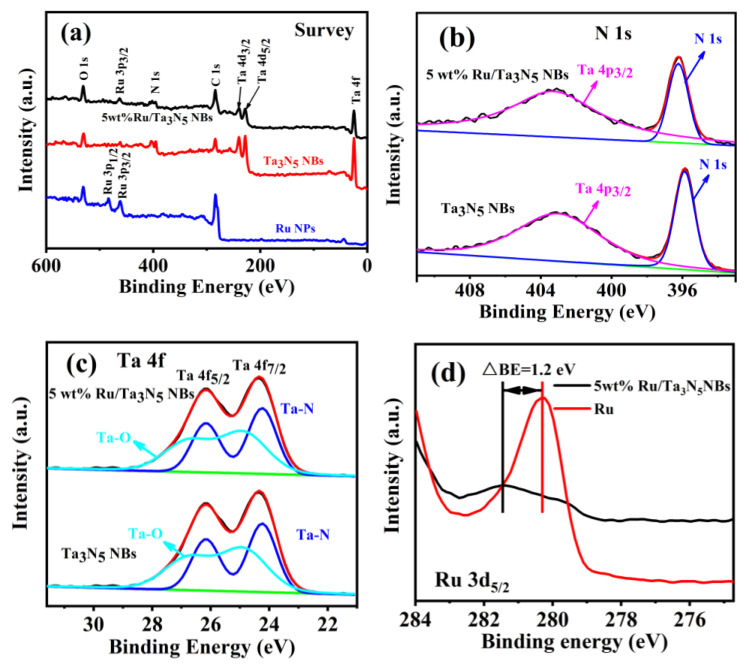
XPS spectra of Ta_3_N_5_, 5 wt% Ru/Ta_3_N_5_, and Ru. (**a**) Survey, (**b**) N 1s, (**c**) Ta 4f, and (**d**) Ru3d_5/2_.

**Figure 5 molecules-28-01100-f005:**
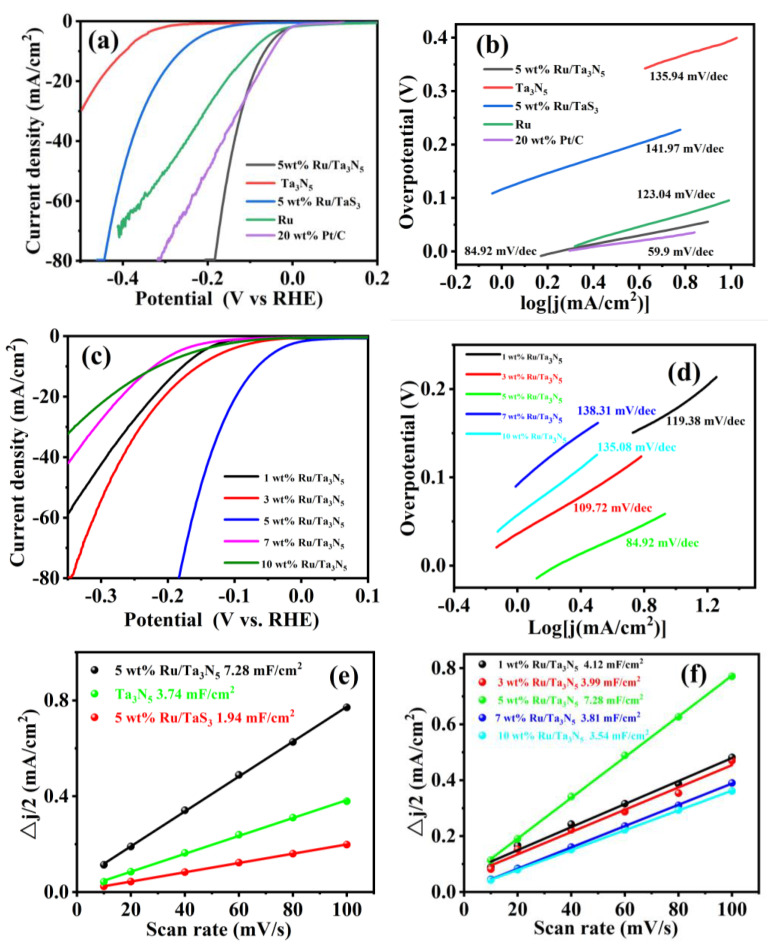
(**a**) LSV and (**b**) Tafel curves of 5 wt% Ru/Ta_3_N_5_ NBs, Ta_3_N_5_ NBs, 5 wt% Ru/TaS_3_ NBs, and Ru (5 wt% Ru/C); (**c**) LSV and (**d**) Tafel curves of 1 wt%, 3 wt%, 5 wt%, 7 wt%, and 10 wt% Ru/Ta_3_N_5_ NBs; (**e**) ΔJ/−v curves of 5 wt% Ru/Ta_3_N_5_ NBs, Ta_3_N_5_ NBs, and 5 wt% Ru/TaS_3_ NBs; (**f**) ΔJ/2−v curves of 1 wt%, 3 wt%, 5 wt%, 7 wt%, and 10 wt% Ru/Ta_3_N_5_ NBs.

**Figure 6 molecules-28-01100-f006:**
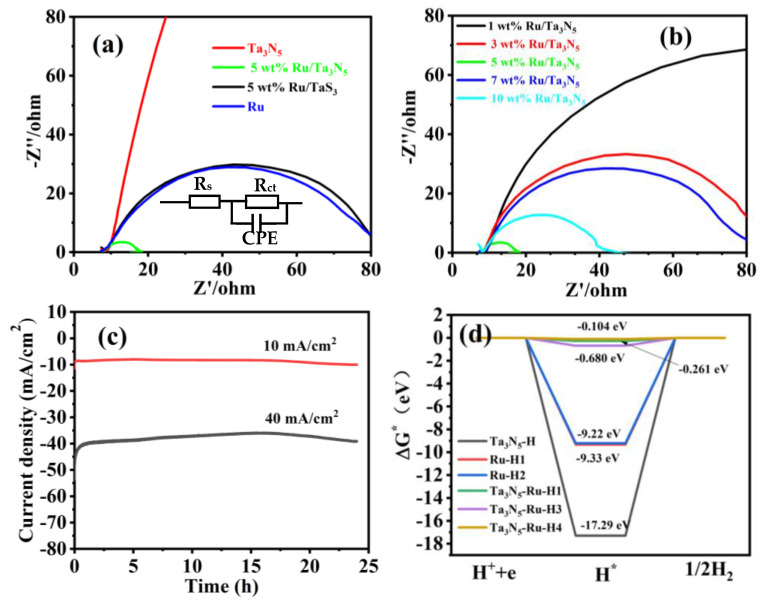
(**a**) EIS curves of 5 wt% Ru/Ta_3_N_5_ NBs, Ta_3_N_5_ NBs, 5 wt% Ru/TaS_3_ NBs, and Ru (5 wt% Ru/C); inset is equivalent circuit. (**b**) EIS curves of 1 wt%, 3 wt%, 5 wt%, 7 wt%, and 10 wt% Ru/Ta_3_N_5_ NBs. (**c**) Chronoamperometric curves of 5 wt% Ru/Ta_3_N_5_ NBs. (**d**) Hydrogen adsorption free energy (ΔG_H*_) on Ta_3_N_5_ (H is adsorbed on Ta_3_N_5_: Ta_3_N_5_−H), Ru (H is adsorbed on different site of Ru: Ru−H1 and Ru−H2), and Ru/Ta_3_N_5_ (H is adsorbed on different site of Ru/Ta_3_N_5_: Ta_3_N_5_−Ru−H1, Ta_3_N_5_−Ru−H2, and Ta_3_N_5_−Ru−H3).

**Figure 7 molecules-28-01100-f007:**
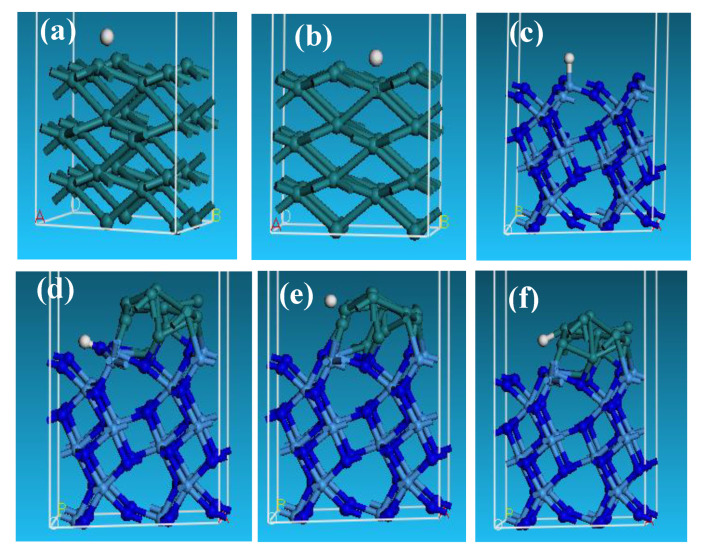
H atom absorbed on various crystalline structures: (**a**) Ru−H1; (**b**) Ru−H2; (**c**) Ta_3_N_5_−H; (**d**) Ta_3_N_5_−Ru−H1; (**e**) Ta_3_N_5_−Ru−H3; (**f**) Ta_3_N_5_−Ru−H4. (H: gray white; Ru: pale green; N: blue; Ta: light gray).

**Table 1 molecules-28-01100-t001:** HER catalytic activity of Ru-loaded hybrids in alkaline electrolytes.

Catalysts	Electrolytes	Overpotential at 10 mAcm^−2^ (mV)	Tafel Slope (mVdec^−1^)	Stability (10 mAcm^−2^)	Ref.
Ru/Ta_3_N_5_ NBs (Ru: 5 wt%)	1.0 M KOH	64.6	84.92	Stability in 24 h	This work
Ru-NMCNs-500 ^1^ (Ru: 3.04 wt%)	1.0 M KOH	28	35.2	Increase of 28 mV after 10,000 cycles	[20]
Ru/MoO_2_/C ^2^	1.0 M KOH	16	32	Stability in 40 h	[21]
Ru-Co@N-Graphene (Ru:3.58 wt%)	1.0 M KOH	28	31	Stability in 10,000 cycles	[23]
B-Ru@CNT (Ru:9.2 wt%; B: 0.06 wt%)	1.0 M KOH	17	30	85% of initial current after 15 h	[24]
Ru/CeO_2_ (Ru:3 wt%)	1.0 M KOH	28.9	53.2	Stability in 18 h	[25]

^1^ Ru single-atom-loading, nitrogen-doped, mesoporous carbon nanospheres annealing at 500 °C; ^2^ Ru: MoO_2_: C is 21.25 wt%: 44.0 wt%: 34.74 wt%.

## Data Availability

No new data were created except for the data in the Appendix A and the paper.

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
