# Peer review of "Ta3N5 Nanobelt-Loaded Ru Nanoparticle Hybrids’ Electrocatalysis for Hydrogen Evolution in Alkaline Media"

_molecules, 2023, doi:10.3390/molecules28031100_

Round 1

Reviewer 1 Report

The manuscript entitled “Ta3N5 Nanobelts-Loaded Ru Nanoparticles Hybrids Electrocatalysis for Hydrogen Evolution Properties in Alkaline Media” describes Ta3N5 nanobelts supported Ru nanoparticles with the Ru loadings ranging from 1 to 10 wt% and evaluated as HER electrocatalyst in alkaline electrolyte. Among them, 5 wt% Ru loaded Ta3N5 nanobelts electrocatalyst affords superior HER activity with a low overpotential of 64.6 mV and Tafel slope of 86.4 mV/dec at 10 mA/cm2 and is also stable up to 25 h. This is an interesting and systematic work. However, there are still several points found in the manuscript requiring further elucidation after my reading in depth.

1.     Why choose Ru loading instead of earth-abundant Ni and Co metal nanoparticles?

2.     The novelty of the work should be established.

3.     Compare your results with the literature ones. Provide a tabular form of the literature on HER performance results with the present work.

4.     Grammar and lots of typological error such as spaces, subscripts, superscripts, etc. is present in the present form of the manuscript. So, need an extensive rectification.

5.     The following references should be added in the introduction part, Tafel, ECSA, and EIS parts: Gong et al. ChemElectroChem. 2019, 6(9):2497-502; Chinese Journal of Catalysis. 2021, 42(8):1387-94; ACS Applied Energy Materials. 2019, 2(10):7256-62.

6.     Generally, chalcogenides are efficient HER active compared to nitrides. In this manuscript, 5wt% Ru/TaS3 has a higher overpotential towards HER compared to 5wt% Ru/Ta3N5. Please explain.

7.     Figure 3: Its looks like microbelts. Provide the size of the material.

8.     Provide elemental analysis (EDS) of the best sample to present “Ru, N, Ta” elements in the sample.

9.     Is it Ru loading or doping? Include more references for validating the advantages of loading with Ru.

Author Response

Answer to report of reviewer 1

The manuscript entitled “Ta3N5 Nanobelts-Loaded Ru Nanoparticles Hybrids Electrocatalysis for Hydrogen Evolution Properties in Alkaline Media” describes Ta3N5 nanobelts supported Ru nanoparticles with the Ru loadings ranging from 1 to 10 wt% and evaluated as HER electrocatalyst in alkaline electrolyte. Among them, 5 wt% Ru loaded Ta3N5 nanobelts electrocatalyst affords superior HER activity with a low overpotential of 64.6 mV and Tafel slope of 86.4 mV/dec at 10 mA/cm2 and is also stable up to 25 h. This is an interesting and systematic work. However, there are still several points found in the manuscript requiring further elucidation after my reading in depth.

  1. Why choose Ru loading instead of earth-abundant Ni and Co metal nanoparticles?

Answer: thanks, this a good idea by using Ni or Co/Ta3N5, and this will be further studied in future work. We once studied Ru/CeO2 for electrocatalysizing water splitting, and hoped to study Ru-loaded catalyst to get a pervasive law, so here we study Ru/Ta3N5 nanobelts for electrocatalysizing water splitting.

  1. The novelty of the work should be established.

Answer:  The novelty of the work has be established. For example, “Ta3N5 nanobelts (NBs)-loaded Ru nanoparticles (NPs) hybrids with various ratios including 1~10 wt% Ru/Ta3N5 are constructed to electrocatalysize splitting water for hydrogen evolution reaction (HER) in alkaline media. The results show that 5 wt% Ru/Ta3N5 NBs have good HER properties with overpotential of 64.6 mV and Tafel slope of 84.92 mV/dec at 10 mA/cm2 in 1 M KOH solution and good stability, and the overpotential of the HER is lower than that of Pt/C (20 wt%) after the current density is 26.3 mA/cm2” and “X-ray photoelectron energy spectroscopy (XPS) demonstrates that the good HER performance comes from the synergistic effect and electronic transfer Ru to Ta3N5. The electrochemical analyses and theoretical calculations exhibit that Ru/Ta3N5 interfaces as real active sites play an important role.” in abstract. It can be included as follows:

  • Ta3N5nanobelts-loaded Ru nanoparticles electrocatalyst are constructed for hydrogen evolution reaction in alkaline media;
  • 5 wt% Ru/Ta3N5nanobelts show good HER properties 1 M KOH solution and good stability;
  • The electrochemical analyses and theoretical calculations exhibitthat Ru/Ta3N5 interfaces as real active sites play an important role.

  1. Compare your results with the literature ones. Provide a tabular form of the literature on HER performance results with the present work.

Answer: A table 1 has been added in text.

Table 1. HER catalytic activity of Ru-loaded hybrids in alkaline electrolytes

Catalysts

Electrolytes

Overpotential at 10 mAcm-2 (mV)

Tafel slope (mVdec-1)

Stability (10 mAcm-2)

Ref.

Ru/Ta3N5 NBs (Ru: 5wt%)

1.0 M KOH

64.6

84.92

Stability in 24 h

This work

Ru-NMCNs-5001

(Ru: 3.04wt% )

1.0 M KOH

28

35.2

Increase of 28 mV after 10000 cycles

20

Ru/MoO2/C2  

1.0 M KOH

16

32

Stability in 40 h

21

Ru-Co@N-Graphene ( Ru:3.58 wt%)

1.0 M KOH

28

31

Stability in 10000 cycles

23

B-Ru@CNT (Ru:9.2 wt%; B: 0.06 wt%)  

1.0 M KOH

17

30

85% of initial current after 15 h

24

Ru/CeO2 (Ru:3 wt%)

1.0 M KOH

28.9

53.2

Stability in 18 h

25

1Ru single atom loading nitrogen doped mesoprous carbon nanospheres annealing at 500 ℃; 2Ru: MoO2: C is 21.25 wt%: 44.0 wt%: 34.74 wt%, respectively.

  1. Grammar and lots of topological error such as spaces, subscripts, superscripts, etc. is present in the present form of the manuscript. So, need an extensive rectification.

Answer: the errors have been rectification. Thanks!

  1. The following references should be added in the introduction part, Tafel, ECSA, and EIS parts: Gong et al. ChemElectroChem. 2019, 6(9):2497-502; Chinese Journal of Catalysis. 2021, 42(8):1387-94; ACS Applied Energy Materials. 2019, 2(10):7256-62.

Answer: These literatures have been cited in introduction as Ref. 6, 7, and 8, respectively.

  1. Generally, chalcogenides are efficient HER active compared to nitrides. In this manuscript, 5wt% Ru/TaS3 has a higher overpotential towards HER compared to 5wt% Ru/Ta3N5. Please explain.

Answer: It is mainly reason that 5 wt% Ru/Ta3N5 has greater electrochemical activity surface area (ESCA) and lower transfer resistances (Rct) than 5wt% Ru/TaS3. A few data and explanations were included in the text.

  1. Figure 3: Its looks like microbelts. Provide the size of the material.

Answer: the sizes of the materials have been provided, which has been written in the paper., for example, “Figure 1(a-d) show the SEM images of TaS3 NBs, Ta3N5 NBs, 5 wt%Ru/Ta3N5 NBs, and 5 wt% Ru/TaS3 NBs, respectively, which present that TaS3 NBs are nanobelts with a width of 1-4 μm, a thickness of about 10-200 nm, and a length up to centimeter, and that the sizes of the Ta3N5 NBs approach those of the TaS3 NBs.”

  1. Provide elemental analysis (EDS) of the best sample to present “Ru, N, Ta” elements in the sample.

Answer: EDS analysis of 5 wt% Ru/Ta3N5 has been shown in Figure 1(f).

  1.   Is it Ru loading or doping? Include more references for validating the advantages of loading with Ru.

Answer: Ru is loaded on Ta3N5 NBs, and references with Ru- loaded electrocatalyst have been cited and discussed in the text. For example, “If Ru is modified, alloyed, or doped, its HER catalytic activity can be improved obviously [20-23]. For example, Sun et al. prepared B-Ru@CNT (carbon nanotubes) electrocatalyst to show excellent HER activity, yielding overpotentials of 17 and 62 mV at a current density of 10 mA cm-2 in alkaline and acidic solutions, respectively. The experimental and theoretical results indicate that the incorporation of B not only weakens the Ru-H bond and downshifts the d-bond centre of Ru from the Fermi level by reducing the electron density at Ru [24]. Similarly, 3 wt% Ru/CeO2 electrocatalyst also demonstrates excellent HER performance, which is also confirmed downshift of the d-band of Ru, and reduction of the Ru-H bond energy [25]. Therefore, Ru is an up and coming material for electrocatalytic HER.” 

Reviewer 2 Report

In this manuscript, Zhang et al. reported the synthesis of Ta3N5 nanobelts-loaded Ru nanoparticles hybrids and their application as alkaline hydrogen evolution reaction (HER) electrocatalysts. Overall, the research was well designed and the results well organized. The conclusions were supported by both experimental and theoretical results. Overall, this work is a good fit for the journal Molecules. Based on the expertise of this reviewer, I would in general support publication. However, some technical issues need to be resolved before the possible acceptance of the manuscript. Please see below for more detail.

1. The writing of the manuscript needs polishing. In particular for the Abstract, some of the technical terms were incorrectly written, including “scan electronic microscopy” (should be “scanning electron microscopy”), “high-solution transmission electronic microscopy” (should be “high-resolution transmission electron microscopy”), and “X-ray photoelectronic energy spectra” (should be “X-ray photoelectron spectroscopy”).

2. While the authors mentioned Ta3N5 in the Introduction, it is suggested that a bit more content be provided to justify why the authors would like to apply Ta3N5 as a support for Ru.

3. The reference section can be improved. (1) Some refs are not relevant to the text and needs replacement. For instance, Ref. 10 is not relevant to the discussion of “sluggish kinetics of alkaline HER”. Double check all other refs. (2) Very recent works on water electrolysis, HER and transition metal oxide HER catalysts are suggested to be referenced (e.g., Energy Technology, 2022, 10, 2200573; Materials Reports: Energy, 2022, DOI: 10.1016/j.matre.2022.100144; Composites Part B: Engineering, 2020, 198, 108214).

4. Because the experimental section is placed near end of the manuscript, the authors are suggested to add some description of the different samples in the beginning of the Results section (section 2.1), so that readers can understand and follow the flow of the manuscript.

5. The Tafel slope of the best sample was claimed to be 86.41 mV/dec in Abstract and Conclusion but was actually 84.92 mV/dec in main text and the relevant figure.

6. Fig 5b and d, their x axis should be revised into the same.

7. Line 115, discussion of Ru 3d XPS result, the bias is given as 1.16 eV but in figure 4d it was 1.2 eV.

8. The abbreviations for hydrogen evolution reaction is inconsistently used, for example, HE in abstract, and HER in main text.

Author Response

Answers to report of reviewer 2

In this manuscript, Zhang et al. reported the synthesis of Ta3N5 nanobelts-loaded Ru nanoparticles hybrids and their application as alkaline hydrogen evolution reaction (HER) electrocatalysts. Overall, the research was well designed and the results well organized. The conclusions were supported by both experimental and theoretical results. Overall, this work is a good fit for the journal Molecules. Based on the expertise of this reviewer, I would in general support publication. However, some technical issues need to be resolved before the possible acceptance of the manuscript. Please see below for more detail.

The writing of the manuscript needs polishing. In particular for the Abstract, some of the technical terms were incorrectly written, including “scan electronic microscopy” (should be “scanning electron microscopy”), “high-solution transmission electronic microscopy” (should be “high-resolution transmission electron microscopy”), and “X-ray photoelectronic energy spectra” (should be “X-ray photoelectron spectroscopy”).

Answer: the errors have been revised, thanks!

While the authors mentioned Ta3N5 in the Introduction, it is suggested that a bit more content be provided to justify why the authors would like to apply Ta3N5 as a support for Ru.

Answer: The question has simply been discussed in introduction. For example, “Ta3N5 is a red powder with an octahedral structure and a narrow band gap of about 2.08 eV, which is of good photosensitive properties [26] and widely applied in solar-driven photocatalytic or photoelectrochemical (PEC) water splitting [27, 28]. Its analogue TaON is also a good photosensitive agent [29] and an ideal photoelectro-catalyst for hydrogen evolution [30, 31]. It brings about a question, if Ru/Ta3N5 is used in electrocatalytical HER under without solar-driven condition, how is its properties of HER?” in introduction, thanks!

The reference section can be improved. (1) Some refs are not relevant to the text and needs replacement. For instance, Ref. 10 is not relevant to the discussion of “sluggish kinetics of alkaline HER”. Double check all other refs. (2) Very recent works on water electrolysis, HER and transition metal oxide HER catalysts are suggested to be referenced (e.g., Energy Technology, 2022, 10, 2200573; Materials Reports: Energy, 2022, DOI: 10.1016/j.matre.2022.100144; Composites Part B: Engineering, 2020, 198, 108214).

Answer: The Ref 10 has been omitted. Energy Technology, 2022, 10, 2200573; Materials Reports: Energy, 2022, DOI: 10.1016/j.matre.2022.100144; Composites Part B: Engineering, 2020, 198, 108214 have been cited as Ref.3, 4, and 5, respectively.

Because the experimental section is placed near end of the manuscript, the authors are suggested to add some description of the different samples in the beginning of the Results section (section 2.1), so that readers can understand and follow the flow of the manuscript.

Answer: The content has been revised as “For preparing Ru/Ta3N5 NBs, firstly TaS3 NBs were synthesized, and then TaS3 NBs were converted to Ta3N5 NBs in the NH3 atmospheres, subsequently, various quantity of Ru3+ was adsorbed on the surface of Ta3N5 NBs by dipping method, finally Ru was loaded on the Ta3N5 NBs by reduction. The structures and morphologies of the catalysts were characterized by X-ray diffractometer (XRD) and scanning electron microscopy (SEM), respectively.”

The Tafel slope of the best sample was claimed to be 86.41 mV/dec in Abstract and Conclusion but was actually 84.92 mV/dec in main text and the relevant figure.

Answer: The 86.41 mV/dec has been revised to 84.92 mV/dec.

Fig 5b and d, their x axis should be revised into the same.

Answer: the X axis units have been revised into the same.

Line 115, discussion of Ru 3d XPS result, the bias is given as 1.16 eV but in figure 4d it was 1.2 eV.

Answer: the bias has been revised to 1.2 eV.Thanks!

  1. The abbreviations for hydrogen evolution reaction is inconsistently used, for example, HE in abstract, and HER in main text.

Answer: HE in abstract has been revised to HER.

Round 2

Reviewer 1 Report

Accept in present form